# Characterizing Pharmacist Perspectives on Utilizing a Personalized Family Medication Safety Plan for Opioid Education with Adolescents and Parents

**DOI:** 10.3390/pharmacy11010022

**Published:** 2023-01-24

**Authors:** Olufunmilola Abraham, Joanne Peters, Kourtney A. Peterson

**Affiliations:** Social and Administrative Sciences Division, University of Wisconsin-Madison School of Pharmacy, Madison, WI 53705, USA

**Keywords:** opioids, adolescents, pharmacist, medication safety, family health

## Abstract

Background: Exposure to prescription opioids during adolescence is associated with an increased risk of future opioid misuse. The pervasive and growing impact of the opioid epidemic requires evidence-based, co-designed interventions targeted at adolescents. MedSMA℞T Families is an intervention tailored to educate adolescents and their families about opioid misuse prevention and consists of two parts: the MedSMA℞T: Adventures in PharmaCity videogame and the family medication safety plan (FMSP). Objective: This study sought to explore pharmacists’ perceptions of using the family medication safety plan to facilitate opioid education among parents and their adolescents. The purpose of this project was to also gather information for iterative adaptations to improve implementation and dissemination of the FMSP in pharmacy settings. Methods: Pharmacists were recruited from Pharmacy Practice Enhancement and Action Research Link (PearlRx) and the Pharmacy Society of Wisconsin (PSW). Twenty-one pharmacist interviews were conducted between September 2021 and March 2022. Consenting pharmacists reviewed the FMSP. Then, semi-structured interviews were conducted, recorded, and transcribed. Inductive thematic analyses were performed using NVivo software. Results: Four prevalent themes emerged: (1) the purpose of FMSP as a communication tool, (2) instructions to clarify how to use FMSP, (3) barriers to using FMSP, and (4) suggestions to improve FMSP format. Most pharmacists described the FMSP as a tool to encourage interactive opioid conversations between adolescents, families, and pharmacists. Pharmacists suggested creating multiple customizable formats and incorporating instructions on how to use the FMSP. Conclusions: Pharmacists noted that the FMSP was an interactive and engaging communication tool to tailor opioid consultations with adolescents and their families. Patients might use the FMSP as a visual cue to help think of what question(s) they should ask pharmacists. Pharmacists stated that the FMSP could facilitate tailored opioid safety communication and medication consultations. Insights will inform future medication misuse prevention interventions as well as adaptation.

## 1. Introduction

In 2020, opioids were involved in more than 68,000 deaths in the United States, which is more than 8.5 times the number of opioid-involved overdose deaths in 1999 [1,2]. Approximately 75% of drug overdose deaths in 2020 involved an opioid [3]. Adolescents are not exempt from the impact of the opioid epidemic. Adolescents that are prescribed opioids before the 12th grade have a 33% increase in the risk of future opioid misuse [4]. Therefore, adolescents represent a vulnerable population since they have limited knowledge about opioid medications and how opioid misuse is defined [5,6,7,8]. A family-driven intervention is essential given the known misuse behaviors that occur within households [9,10,11]. For example, parents often keep leftover opioids and model inappropriate prescription opioid use by sharing unused medications with their children to treat minor injuries. In addition, adolescents may take medications from home medicine cabinets, or parents may give children incorrect dosages. Accordingly, the literature suggests that educating adolescents and their families about safe opioid use, storage, and disposal practices has the potential to reduce opioid misuse. Prior studies demonstrate the receptiveness to and efficacy of pharmacist-led drug consultations [12,13]. As one of the most easily accessible health care professionals, pharmacists are uniquely positioned to educate patients and their families about safe and appropriate prescription opioid use.

Pharmacists primarily rely on verbal consultations and medication information sheets to educate both adult and pediatric patients. Adult and pediatric patients alike struggle with the high level of health literacy required to interpret and understand medical information sheets. Further, patients are often hesitant to ask questions during verbal consultations because they are unsure of what to ask [12]. This results in medication consultations that involve one-way communication, where the pharmacist provides medication information and patients passively listen. Ideally, pharmacist–patient medication consultations should involve patient engagement and active participation. Unfortunately, pharmacists have limited education tools to engage parents and adolescent patients during consultations, and education tools tailored to engage pediatric patients are even more scarce [14,15].

To bridge this gap, the MedSMA℞T Families intervention was designed and tailored to educate adolescents and their families on safe and appropriate opioid use. The MedSMA℞T Families intervention uses two parts: (1) the MedSMA℞T: Adventures in PharmaCity game and (2) the family medication safety plan (FMSP).

The FMSP (Appendix A: Figure A1) is a tool that was designed to capture vital opioid medication information that is crucial for safe and appropriate prescription opioid use as well as encourage communication between pharmacists, adolescents, and families. Currently, there are no standard ways to encourage parents to have medication safety conversations with their children. Many of the available education programs regarding medication management and safety are targeted towards schools (e.g., Rx for Addiction and Medication Safety) [16]. The available patient medication safety plans are not focused on opioid management and lack scientifically based foundations. The FMSP was developed by the research team engaging with over 60 parent and adolescent dyads in creating this tool. The parent and adolescent dyads were asked about current medication management in their homes as well as learnings from the MedSMA℞T game which then led to creating their own medication safety plan. The FMSP was designed to help families integrate and put into practice what they have learned from the serious game. The research team’s 12-member Youth Advisory Board also reviewed original drafts of the FMSP and offered suggestions for improvement to facilitate use by adolescents and their parents.

The FMSP has five sections: (1) medication and family information, (2) dosage and instructions, (3) medication schedule, (4) proper storage and disposal, and (5) positive communication. The objective of this manuscript was to characterize pharmacists’ perceptions of the FMSP and inform future iterations of the tool for use in pharmacy practice. MedSMA℞T: Adventures in PharmaCity is a serious game (Appendix B: Figure A2) that teaches important components of opioid safety that are reinforced in the FMSP [12,13,14].

## 2. Materials and Methods

### 2.1. Study Design

This was a cross-sectional, qualitative study that utilized virtual, semi-structured interviews via Webex with pharmacists. Data were collected between September 2021 and March 2022. The Institutional Review Board (IRB) at the University of Wisconsin-Madison granted approval for this research prior to the start of the study.

### 2.2. Participants and Recruitment

Eligible participants were pharmacists with access to a computer with webcam. Pharmacists were recruited with email listservs from the Pharmacy Practice Enhancement and Action Research Link (PearlRx) and the Pharmacy Society of Wisconsin (PSW). There were no other exclusion criteria. These venues allowed for the participation of pharmacists throughout the state of Wisconsin. Interested pharmacists were instructed to contact the study group directly via email. The study team followed-up with pharmacists no more than three times via email to schedule a virtual (Webex) session. Participants were able to set up the study session at a time of their choosing.

### 2.3. Data Collection

During the study session, pharmacists provided information on demographic characteristics via a Qualtrics survey, reviewed the Family Medication Safety Plan (FMSP), and participated in semi-structured interviews, which were approximately 45 min long. Pharmacists received an Amazon gift card for their participation. The research team member that interviewed study participants documented detailed reflection notes after each interview. The study team created an interview guide, which asked pharmacists about their perceptions of the FMSP. Interview questions included: the purpose of the FMSP, how the FMSP can impact pediatric medication safety, facilitators, and barriers to the implementation of the FMSP in a pharmacy practice setting, perceived usefulness of the FMSP, and suggestions for improvements. A complete list of questions from the interview guide is presented in Appendix C. Interviews were audio-recorded via Webex. Audio-recorded interviews were professionally transcribed, verbatim. Recordings were stored securely on the restricted drive and de-identified transcripts were stored in Box.

### 2.4. Data Analyses

Two members of the research team (coders) independently coded each transcript using a thematic analysis approach. Coders (JP, KP) used the software NVivo for coding all interview transcripts. Coders began by using an inductive semantic approach to generate codes relevant to pharmacists’ opinions and perceptions of the FMSP. These codes were sorted into themes by the coders finding patterns and repetition in the data. The coders met regularly to discuss the codes and address differences, later they met to revise the master codebook [17,18,19,20]. Results were selected based on the prevalence of emergent themes across all transcripts and their relevance to pharmacist perceptions, adaptation, dissemination, and implementation of the FMSP in pharmacies. Thematic analysis has been used in health services research to characterize the perspectives of patients and health care providers alike. Thematic analysis allows for flexibility through inductive reasoning, allowing for themes to arise from the data. This flexibility makes it especially appealing to research concerned with dissemination and implementation as it can be used to characterize unique facilitators and barriers in each context.

Descriptive statistics of pharmacists’ demographics were generated and are reported below to characterize and contextualize the study population in relation to the results [21,22,23].

## 3. Results

Interviews were conducted with 21 pharmacists, and 20 of them reported demographic information. The average age for pharmacists was 35 years old and most identified as female (80%) and white (90%). Pharmacists commonly worked in inpatient pharmacies (35%) or independent pharmacies (25%). Accordingly, 60% of respondents identified as either clinical hospital (35%) or community pharmacists (25%). Other pharmacy specialties included managed care, population health, and pediatric in-patient. Pharmacists spent 11 years in pharmacy practice and 5.5 years in their current practice setting on average. Among pharmacists who filled prescriptions, approximately 1383 prescriptions were filled per week on average. Pharmacist demographics are summarized in Table A1 (Appendix D).

Four main themes emerged: (1) purpose of FMSP as a communication tool, (2) instructions to clarify how to use FMSP, (3) barriers to using FMSP, and (4) suggestions to improve FMSP format. Table A2 (Appendix E) provides a summary of the themes, subthemes, and verbatim quotes from study participants. Themes and subthemes are described in detail below. Complete verbatim quotes by subtheme are listed in Table A3 (Appendix F).

### 3.1. Theme 1: Purpose of FMSP as a Communication Tool

#### 3.1.1. Facilitate Patient-Pharmacist Communication and Encourage Patient Questions

All pharmacists identified the purpose of the FMSP as a tool to facilitate patient-pharmacist communication. Most pharmacists suggested that the FMSP could encourage patients to ask questions, making the consultation more interactive and engaging for both parties. Pharmacists suggested that the FMSP could be used for other groups in addition to adolescent patients with prescribed opioids, such as patients with low health literacy, older adults, large families, new patients, patients taking multiple medications, patients who incorrectly use their medication, patients on short term medication therapy, and high-risk medications (including stimulants and benzodiazepines). The positive communication section was where patients could write down questions or concerns to discuss with their pharmacist or doctor. A few pharmacists suggested that including some frequently asked patient questions would help clarify what is meant by “positive communication” and some suggested including answers to commonly asked questions as well for patients to reference.


*“Like I said, it actually makes them [patients] think about their medicines and where they can communicate, who they can communicate with if they have questions, which is nice, because I think some people don’t understand that or think about that at all.”*
—Pharmacist 7


*“The FMSP is interactive because it’s not prefilled out, and it does encourage a discussion, which can then prompt questions rather than just reading a bunch of stuff on a page.”*
—Pharmacist 21


*“And then I guess I would maybe think about how we could frame this positive communication section to like just not be an empty box. Maybe like what are the most common questions that patients and families have and include that, because I just, I’m not sure what people would put in there. Maybe I’m not understanding the intent, but I’m not entirely sure like I would know what to write in there.”*
—Pharmacist 20

#### 3.1.2. Increase Patient Awareness, Education, and Safety

Pharmacists highlighted that the FMSP has the potential to increase opioid awareness, education, and safety among patients and decrease the stigma around opioid medications The FMSP included important opioid medication information, e.g., concerning how to safely use opioids, as well as appropriate opioid storage and disposal. Many pharmacists emphasized that they liked the opioid storage and disposal information included in the FMSP. Verbal medication consultations frequently include information about the purpose of the medication (i.e., why the patient is prescribed the opioid), how to take the medication, and what to expect from the medication (i.e., side effects and benefits). However, topics, such as opioid storage and disposal, are discussed less frequently during medication consultation, which is often due to time constraints due to limited patient consultation time. Time constraints force pharmacists to be concise and prioritize what they think is important for patients to know during the consultation. Unfortunately, safe opioid storage and disposal are usually considered lower priority topics and thus are frequently omitted from the discussion. However, having a tool like the FMSP could remind pharmacists to discuss appropriate opioid storage and disposal. Moreover, the FMSP could serve as a tool to remind patients to ask questions about appropriate opioid storage and disposal if it is not explicitly discussed during the consultation.


*“So I think just opening the line of communication, which then I think increases safety right there by being a communication bridge between family and patients.”*
—Pharmacist 8


*“I think the goal of it is probably to help children, families, and patients have an appropriate idea of how to safely use their medications at home and what to do when they’re done with them. Honestly, the document maybe isn’t as important as the conversation, but the document facilitates that and then also starts to maybe change people’s impressions of how medications should be handled. In my mind at least, I envision this being a facilitator of a conversation that happens at counseling.”*
—Pharmacist 13

### 3.2. Theme 2: Instructions to Clarify How to Use FMSP

#### 3.2.1. Person Who Completes the Information in the FMSP

One of the most common questions pharmacists had about the clarity of the FMSP concerned “How should it be used and who should be filling out the information in the form?” Pharmacists had multiple suggestions for ways it could be used during a consultation but wanted instructions regarding how it should be used. In the study’s format of the FMSP there are no explicit instructions regarding who should fill out the information (whether it should be the child, parent, or pharmacist). This was a purposeful decision to not include who should fill out the FMSP so as not to restrict potential ideas regarding how it could be used or incorporated into practice. Pharmacists mentioned that if patients were the ones to fill out the FMSP on their own, they might not understand certain sections, such as drug–drug and drug–food interactions or potential side effects. If patients are expected to complete the FMSP on their own, not knowing what to include in which section could be a barrier to completion. Other suggestions that pharmacists had on who should fill out the FMSP included nurses, pharmacists, pharmacy technicians, other healthcare professionals, and caregivers.


*“And am I, as a pharmacist, supposed to go through this with the patient, or is the patient supposed to be filling in their drug-food interactions? How would they ever know that information? This would just be super overwhelming. And I think, if I was the patients, I would just quit.”*
—Pharmacist 16

#### 3.2.2. Medications to Include in the FMSP

Many pharmacists also asked if the intent of the FMSP was to include all the medications that the patient is currently on or just include opioids and other medications for pain relief. Which medications to include has implications for how to complete other sections of the FMSP, such as drug interactions, side effects, and what to do in the case of accidental overdose, as well the potential for patients to forget to take medications not included on the schedule.


*“It’s not entirely clear that this plan is intended for patients who are on opioids. It looks like it could be used for anybody who’s taking multiple medications. So those things are still helpful, but I guess what to do in case of accidental use or overdose may not always be as relevant.”*
—Pharmacist 9

### 3.3. Theme 3: Barriers to Using FMSP

#### 3.3.1. Time and Setting

When asked about barriers to using the FMSP in practice, most pharmacists mentioned time and the practice setting. Both pharmacist and patient time constraints were noted. The pharmacy setting can create time constraints. In a retail pharmacy setting, there may not be as much time to provide medication consultations as there could be for in-patient or appointment-based encounters.


*“Probably time. I mean, it would take time to go through this with the pharmacist or to fill it out yourself.”*
—Pharmacist 10


*“It depends on the counseling. If it’s appointment-based counseling, then I think we have plenty of time and resources. But if it’s more so just drop-in counseling, I don’t think patients will have the time to do it, and it would disrupt our workflow.”*
—Pharmacist 2

#### 3.3.2. Patient-Related Barriers

Pharmacists commented on potential patient-related barriers, which included: patients on more than three medications, patients who speak another language, and patients with multiple care takers. Pharmacists stated that patients who are on more than three medications might not have adequate space to include all their current medications in the FMSP. This is true for older adults or patients with chronic medical conditions, who are prescribed complex regimens involving numerous medications. Similarly, language barriers might exist for patients who primarily speak another language besides English. One suggestion for overcoming this language barrier would be to include less written text and incorporate more pictures. Pictograms can be used to get the same information across without requiring the patient to be literate in a specific language and can also be beneficial in engaging younger children. 

Another common suggestion was to have the FMSP translated into languages commonly spoken in the locale of the pharmacy. Patients with multiple care takers could be a barrier to the efficient use of the FMSP depending on the format. For example, if the format of the FMSP was on a single sheet of paper, it could be lost or left in a place (such as school or at one parent’s house), making it unavailable to use by other care takers who do not have access to that physical sheet of paper. 


*“But I think it would be more challenging for people that take a lot of medications because it just would require a lot of upkeep, and medications change a lot.”*
—Pharmacist 9


*“Yeah, so I think maybe use it making sure to use the simplest language possible, offering it in other languages if needed, if English isn’t the first language. Maybe using pictures where you can.”*
—Pharmacist 11


*“I think it’s a good communication tool to keep everyone in the family informed of what’s going on because, in general, only one person is usually going to pick up the medication. It’s not going to be a whole family affair. You might have parents and older siblings and grandparents who are all helping the child or adolescent, and it’s nice to have everyone filled in.”*
—Pharmacist 11

#### 3.3.3. Patient-Pharmacist Buy-in

Pharmacists stated that buy-in from the pharmacist, pharmacy site managers, as well as the patient would be required to successfully incorporate the FMSP into practice. For the FMSP to be used consistently, both patient and pharmacist stakeholders need to think the FMSP provides additional value as compared to the current status quo. If the patient does not see value in completing the FMSP, it is unlikely that they will complete the form or reference the completed form. Similarly, if the pharmacists see no relative advantage of the FMSP compared to current patient education tools, they are unlikely to consistently incorporate it into practice or provide motivational consultation when using the FMSP.


*“And maybe patient buy-in too, because it, depending on the practice setting, I think patients or family members might not expect this of their pharmacist.”*
—Pharmacist 14


*“I think its staff buy-in because something like this [the FMSP] isn’t going to work unless you consistently do it, but that means everybody on the team has to be on board. And a lot of times, that can be kind of hard.”*
—Pharmacist 13

### 3.4. Theme 4: Suggestions to Improve FMSP Format

#### Multiple Modifiable Formats Tailored to Patient Preferences

Pharmacists suggested creating multiple modifiable formats (paper, electronic, smartphone app, etc.) tailored to patient and family preferences. Most pharmacists agreed that a single format for the FMSP could have significant limitations for specific populations. For example, paper and pencil might be easier for patients without access to a smartphone, but a single sheet of paper could be easily lost or kept in a location where it can only be accessed at certain times, e.g., leaving it on the refrigerator at home. An electronic format of the FMSP would allow patients to have it with them at all times on their smartphones. However, this would require patients to have (or somehow get access to) smartphones. Pharmacists mentioned incorporating the FMSP into the patient’s electronic health record so providers have access to it and can work with the patient to edit it as needed.


*“I think ideally, it’s a combination of them [paper and electronic formats]. I feel like it might be more easily accessible if it was electronic, like through an app so that I could access wherever I am. But on the other hand, there might be benefits of having it on paper because I could update it. You could easily fill it out without a smartphone and have access to it. So, I was thinking of multiple applications just based on different patients’ preferences.”*
—Pharmacist 17

## 4. Discussion

Pharmacists recognized the importance of the FMSP as a communication tool to encourage patients to ask questions to make consultation more interactive and engaging. Prior research has indicated the lack of pharmacist-child consultation in community pharmacies, highlighting the importance of the FMSP in facilitating pharmacists in discussing important medication details with both parents and children [24]. However, pharmacists utilizing this tailored approach may face barriers, such as time constraints and the absence of children in community pharmacies [13]. Literature has indicated that when a patient asks a single open-ended question, the pharmacist is significantly more likely to provide additional medication information in response to that question [25]. Thus, it is reasonable to posit that when a pharmacist asks an open-ended question, the patient is significantly more likely to provide more information in response. The ability of the FMSP to act as a discussion guide for patients and pharmacists is a welcome intervention that could provide patients with a more robust understanding of their medication and medication safety practices. 

The FMSP is an important tool to encourage patients to think about what to ask their pharmacist. Parents often misunderstand or are unaware (having never been told) of safe disposal methods for opioid medication. One study found that only 11.7% of parents with older children (7–17 years old) and 29.0% with younger and older children (1–17 years old) in the home knew how to safely store these medications [26]. While another study in pediatric emergency rooms found that more than 60% of caregivers of children 10 years old and older intended to keep their left-over prescription opioids or have, already, intentionally kept left over prescription opioids in the home. Over half of those caregivers intended to keep the prescription for the patient in the future, while 5.9–9.5% intended to keep the medication for someone else. Of the study’s participants, 7.1% reported they would give left over opioids to their adolescents for pain management [27]. These studies indicate that the FMSP may provide both patients and pharmacists with the needed reminder to discuss these issues. The FMSP fills the need expressed by pharmacists for tools designed to facilitate conversation regarding opioid safety.

Recent meta-analyses and studies have also demonstrated positive impacts of pharmacist-led, educational interventions on medication adherence and other outcomes [28,29]. The current study demonstrates the interest in and reported usefulness of a novel intervention for pharmacists to use in verbal medication consultation with their patients. Pharmacists advocated for including instructions with the FMSP. The research team purposely did not include instructions to prevent introducing potential biases into the pharmacists’ suggestions for how the FMSP could be used. One such type of instruction pharmacists discussed was explicit instruction on which medications (pain medications only or all current patient medications) to include. Including only pain medications in the FMSP would be faster to complete and allows for a more targeted consultation. Conversely, including only pain medications could have medication safety implications for patients who are prescribed benzodiazepines, and may also result in drug-drug interactions not being assessed properly.

Feedback from pharmacists suggests that instructions were needed to clarify roles and increase usability. Clarifying roles in the FMSP (adolescent, parent, or pharmacist) is necessary to reduce role confusion [30]. When roles are explicitly defined, it can empower individuals and encourage self-efficacy because then they know what is expected of them. Without clear expectations, role ambiguity can act as a significant barrier. Additional research on adolescents’ and families’ perspectives are necessary to better understand what they want their role to be in completing the FMSP so it can be defined in the instructions.

Our previous research has characterized adolescent preferences for opioid education and what topics are most important to them [31]. However, pilot testing with youth and their caregivers will be crucial to determine how adolescent and family role preferences can be reconciled with existing pharmacist workflows. This study found time constraints to be a significant potential barrier. Future iterations of the FMSP will allow for modulable formats to adapt to time constraints challenges in pharmacies. By making the FMSP amendable to pharmacy time constraints it may increase buy-in for pharmacists and therefore parents and adolescents. Other patient-related factors (language, multiple medications, and multiple care takers) pose additional complex, practical barriers. Nevertheless, leveraging technology might have the potential to help with some of these barriers. For example, technology can be used to help translate the FMSP into other languages. However, a native speaker of the language would need to read the translated text and make sure it is culturally relevant. 

Most pharmacists commented on the FMSP format, which is arguably the most complex and important decision that impacts usability. If stakeholders find the FMSP difficult to use, they might think the intervention is more work than it is worth. Many pharmacists stated that having multiple formats might work best. However, creating multiple formats of the FMSP requires time and resources. Electronic formats such as an app might require help from an informational technology (IT) specialist to create and sustain the app, which can be expensive. Making a modifiable electronic format sounds optimal but would require significant user education and on-going IT support to troubleshoot user issues as they arise. Previous research has demonstrated the acceptability of digital-based applications for medication adherence, indicating that future iterations of the FMSP should have a digital option [30]. The FMSP could be integrated into a patient’s electronic healthcare record with similar barriers. This will take time and resources, but it could greatly impact usability, successful implementation, and sustainability, and slow or limit intervention drift.

### Limitations

Results must be generalized prudently. Since interviews were recorded via Webex, pharmacists may have been more likely to answer interview questions in a socially desirable fashion. If this occurred, pharmacists would likely over-emphasize the benefits and minimize the limitations of the FMSP. Nevertheless, pharmacists were explicitly told that the answers they give to interview questions were confidential, deidentified, and had no impact on study compensation. This research used a convenience sample of Wisconsin pharmacists living in close geographic proximity. Hence, the sample of pharmacists was relatively homogenous in terms of gender, race, and ethnicity. Notwithstanding, interviews were conducted until data saturation was reached, which indicated a sufficient sample size. 

## 5. Conclusions

Pharmacists play an essential role in medication safety by educating patients about the medication, what the medication is for, how to use the medication, what to expect while taking the medication (i.e., side effects), and answer patient questions or concerns. Patients might use the FMSP as a visual cue to help think of what question(s) they should ask. Many pharmacists stated that the FMSP could facilitate tailored opioid safety communication and could engage patients and families during medication consultations. Although the FMSP was created for families, pharmacists agreed that the FMSP could be useful to many other patient populations including patients with lower health literacy, older adults who are frequently on complex medication regimens, care givers, etc. Careful consideration is needed to refine the FMSP and make it more patient friendly while still being concise. Including pictograms could be particularly beneficial strategies mentioned by pharmacists to get ideas across without relying on the patient’s health literacy and/or language skills. 

Pilot testing with a larger and increasingly diverse patient population in different pharmacies and other clinical healthcare settings will help to refine the FMSP. These refinements to the FMSP will likely depend upon the clinical practice setting, which ultimately could require creating multiple FMSP formats. Continuous iterative feedback from pharmacists, adolescents, and parents will be essential in making appropriate adaptations to the FMSP. Future iterations may be explored by including other medication classes in the FMSP. Further research could characterize the needs and wants of other patient populations with different medications. The next steps include the in-situ study of time constraints and role preferences, adapting the FMSP to contain instructions, and designing a FMSP with a modifiable format and modifications to ensure it is accessible and understandable by the lay patient population.

## Data Availability

The data presented in this study are available on request from the corresponding author. The data are not publicly available due to protecting privacy of study participants and ethical considerations.

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
