# Peer review of "Characterizing Pharmacist Perspectives on Utilizing a Personalized Family Medication Safety Plan for Opioid Education with Adolescents and Parents"

_pharmacy, 2023, doi:10.3390/pharmacy11010022_

Round 1

Reviewer 1 Report

In this study, interviews were conducted with pharmacists to understand their perceptions of using the family medication safety plan (FMSP) to communicate with adolescents and their parents about opioids. IRB approval was obtained before the study began. I appreciate the authors undertaking this study and the inclusion of the FMSP and interview guide as appendices to the manuscript. Noted strengths are the study addresses the relevant topics of safe opioid use and prevention of opioid misuse and the manuscript is clear and concise. I noted only minor weaknesses that can be addressed by the authors through editing the manuscript.

Major comments:

None

Minor comments:

Introduction

Has a description of how the FMSP was developed been published previously? If yes, please reference so readers can consult for more detail. If no, please provide a description of how the FMSP was developed, either in the introduction or methods section. Are there tools already in existence on which the FSMP is based? If yes, please reference and provide the relative advantages of the FSMP. It will be helpful for readers to understand how the FMSP for this study came to exist in its current form.

·         The FMSP does not appear to be limited to use with opioids, as it looks like it could be used for any medication. Has the FMSP been used with non-opioid medications? If yes, please provide references.  

Materials and Methods

In section 2.2, please provide more detail about how pharmacists were recruited for the study and which pharmacists were targeted. Also, please note whether pharmacists had to self-identify as providing opioid counseling/education to adolescents/parents to be eligible. Were there any other eligibility criteria? These edits will help readers better understand who participated in the interviews.

Please include a statement of compensation provided to participants in main text. I only see mention of an Amazon gift card in Appendix B.

In section 2.3, please provide detail about who conducted the interviews.

In section 2.4, line 105: please include initials of the two team members who coded the transcripts

In section 2.4, line 108: the statement “Results were selected based on prevalence of emergent themes.” is unclear to me. Please add more detail to make clearer.

Results

Appendix C/Table 1: not currently in table form. Perhaps this issue was created when the file was uploaded? Please edit.

·         Lines 123 and 124: Please only refer to as Appendix C in the main text. Referring to it as Table 1 in the main text gives the impression there is a Table 1 in the main text.

Lines 158 and 159: while I agree the FMSP contains many pieces of information about a medication, I don’t agree with this phrase “…such as what opioids are…”. I don’t see a field on the FMSP that explains what an opioid is. Please clarify.

Subheading for Theme 3 (line 216): would read better to state as Barriers to using FMSP. Please edit in abstract (line 24), results (line 122), and Appendix C (line 574) as well.

Subheading for Theme 4 (line 280): would read better to state as Suggestions to improve FMSP format.

Appendix C, line 620: Pharmacist 1 stated “In the survey, I know it asked something about like how confident are you…”. The methods section indicates the Qualtrics survey collected demographic information but does not indicate other data were collected. It’s certainly possible the Qualtrics survey collected data that are not included in the manuscript because they were not within scope, and that’s fine. In that case, including Pharmacist 1’s comment creates confusion. Please rectify this discrepancy.

Discussion

Line 323: did you intend to write “metanalyses” or “meta-analyses”?

Lines 326-329: it is not clear to me why the efficacy of previous pharmacist-led interventions demonstrates that pharmacist concerns about patient-pharmacist buy-in are important for implementation. Please clarify.

Sentence that spans lines 352-354: the meaning of this sentence is not clear to me. Please clarify.

Sentence that spans lines 357-358: the meaning of this sentence is not clear to me. Please clarify.

Sentence that spans lines 373-375: the flow of this sentence is not smooth. Perhaps “and” between “successful implementation” and “sustainability” is needed? Please edit.

Sentences spanning lines 392-400: typos in these sentences make them more difficult to comprehend. Please edit.

Sentence spanning lines 410-414: typos in this sentence make it more difficult to comprehend. Please edit.

References

References 7 and 17 are identical. Please remove duplicate reference.

Author Response

Reviewer 1 Feedback

Team Response

Has a description of how the FMSP was developed been published previously? If yes, please reference so readers can consult for more detail. If no, please provide a description of how the FMSP was developed, either in the introduction or methods section.

Thank you for the feedback. No, however we have only published on the game previously. We have added information about the development of the FMSP.

. Are there tools already in existence on which the FSMP is based? If yes, please reference and provide the relative advantages of the FSMP. It will be helpful for readers to understand how the FMSP for this study came to exist in its current form.

Thank you for your feedback. We have added information about how the FMSP came to be and what other tools exist presently.

·         The FMSP does not appear to be limited to use with opioids, as it looks like it could be used for any medication. Has the FMSP been used with non-opioid medications? If yes, please provide references. 

We appreciate your question. No, it has not been used with non-opioid medications.

In section 2.2, please provide more detail about how pharmacists were recruited for the study and which pharmacists were targeted. Also, please note whether pharmacists had to self-identify as providing opioid counseling/education to adolescents/parents to be eligible. Were there any other eligibility criteria? These edits will help readers better understand who participated in the interviews.

Thank you for your comments and questions. We have added additional information.

Please include a statement of compensation provided to participants in main text. I only see mention of an Amazon gift card in Appendix B.

Thank you. We have included this information in the Data Collection section.

In section 2.3, please provide detail about who conducted the interviews.

We appreciate your feedback; we have added information regarding this.

In section 2.4, line 105: please include initials of the two team members who coded the transcripts

Thank you, we have included their initials.

In section 2.4, line 108: the statement “Results were selected based on prevalence of emergent themes.” is unclear to me. Please add more detail to make clearer.

Thank you, we have expanded and made this clearer.

Appendix C/Table 1: not currently in table form. Perhaps this issue was created when the file was uploaded? Please edit.

Thank you, this is now formatted as a table.

Lines 123 and 124: Please only refer to as Appendix C in the main text. Referring to it as Table 1 in the main text gives the impression there is a Table 1 in the main text.

Thank you for this input, we have changed accordingly.

Lines 158 and 159: while I agree the FMSP contains many pieces of information about a medication, I don’t agree with this phrase “…such as what opioids are…”. I don’t see a field on the FMSP that explains what an opioid is. Please clarify.

Thank you, we have removed this from the manuscript.

Subheading for Theme 3 (line 216): would read better to state as Barriers to using FMSP. Please edit in abstract (line 24), results (line 122), and Appendix C (line 574) as well.

Thank you for the suggestion, we have made this change.

Subheading for Theme 4 (line 280): would read better to state as Suggestions to improve FMSP format.

Thank you, we have changed it.

Appendix C, line 620: Pharmacist 1 stated “In the survey, I know it asked something about like how confident are you…”. The methods section indicates the Qualtrics survey collected demographic information but does not indicate other data were collected. It’s certainly possible the Qualtrics survey collected data that are not included in the manuscript because they were not within scope, and that’s fine. In that case, including Pharmacist 1’s comment creates confusion. Please rectify this discrepancy.

Thank you; our apologies for the confusion. We have made an edit to this to not create additional confusion.

Line 323: did you intend to write “metanalyses” or “meta-analyses”?

Thank you, we have corrected this typo.

Lines 326-329: it is not clear to me why the efficacy of previous pharmacist-led interventions demonstrates that pharmacist concerns about patient-pharmacist buy-in are important for implementation. Please clarify.

We appreciate the response; we have removed this section for clarity.

Sentence that spans lines 352-354: the meaning of this sentence is not clear to me. Please clarify.

Thank you, we have edited this section for clarity.

Sentence that spans lines 357-358: the meaning of this sentence is not clear to me. Please clarify.

We appreciate the response; we have made corrections for clarity.

Sentences spanning lines 392-400: typos in these sentences make them more difficult to comprehend. Please edit.

Thank you, we have addressed the typos.

Sentence spanning lines 410-414: typos in this sentence make it more difficult to comprehend. Please edit.

Thank you, we have corrected typos.

References 7 and 17 are identical. Please remove duplicate reference.

Thank you, we have fixed this issue.

Reviewer 2 Report

To make the paper complete, the reviewer recommends expanding the introductory part a bit. The reviewer recommends adding some reference literature

A reviewer's advice is to create some tables or graphs in the results section to make the results more intuitive and immediate

Author Response

Reviewer 2 Comments

Team Response

To make the paper complete, the reviewer recommends expanding the introductory part a bit. The reviewer recommends adding some reference literature.

Thank you. We have made additions to the introduction with relevant literature.

A reviewer's advice is to create some tables or graphs in the results section to make the results more intuitive and immediate.

Thank you for the feedback; we have created additional tables.
